# Real-Time and Long-Term Monitoring of Coastal Water Turbidity Using an Ocean Buoy Equipped with an ADCP

**DOI:** 10.3390/s24216979

**Published:** 2024-10-30

**Authors:** Jia-Wei Bian, Ching-Jer Huang

**Affiliations:** 1Department of Hydraulic and Ocean Engineering, National Cheng Kung University, Tainan 70101, Taiwan; 2Coastal Ocean Monitoring Center, National Cheng Kung University, Tainan 70101, Taiwan

**Keywords:** ADCP, echo intensity, turbidity, suspended sediment concentration, ocean data buoy, real-time monitoring

## Abstract

In this study, an acoustic Doppler current profiler (ADCP) operating at 600 kHz was installed on an ocean data buoy in the Qigu waters, Taiwan, to gather real-time sound echo intensity data. These data were then correlated with turbidity measurements obtained by a turbidimeter mounted on the buoy’s mooring line at a water depth of 13 m. The data buoy operated from 6 June to 16 August 2017. During this period, turbidity measurements were recorded from 6 to 21 June 2017. This study established a calibration between the sound echo intensity measured by the ADCP and the turbidity measured using the turbidimeter; a strong linear correlation was discovered between these two variables. This correlation enabled the conversion of echo intensity data into a continuous time series of turbidity measurements, facilitating real-time and long-term monitoring of coastal water turbidity through the deployment of a buoy equipped with an ADCP. The relationships between turbidity and environmental factors such as rainfall, tides, current speeds, and wave activity over an extended period were then investigated. The results revealed that stronger tides and currents in the Qigu waters often lead to higher turbidity, suggesting that these two factors are the primary driving forces for sediment transport in the Qigu waters. Additionally, sampling of water in the Qigu area revealed sediment particles of size ranging from 2 to 120 μm.

## 1. Introduction

The real-time and long-term measurement of suspended sediment concentration (SSC) or turbidity in coastal waters provides essential data for understanding coastal dynamics and variations in water quality [1].

Traditional methods for measuring SSC or turbidity include direct sampling and filtering and optical methods [2,3,4]. However, these methods have their limitations. Direct sampling may yield inaccurate results due to spatial differences in sampling and subsequent laboratory analysis. Optical methods are difficult to maintain over a long period in the underwater environment because the sensors can be disrupted by the presence of biological organisms [5].

Recent advancements have introduced time-domain reflectometry and laser diffraction techniques for measuring SSC in water. In time-domain reflectometry, an instrument is placed on the water’s surface that emits electromagnetic waves through a waveguide submerged beneath the surface. The reflected signals are then analyzed. This technique is feasible due to the substantial contrast in dielectric properties between water and sediment [6,7]. Nevertheless, the strong absorption of electromagnetic waves in seawater limits the application of this sensor in marine environments. Laser diffraction analysis involves projecting a laser beam onto a small underwater area and rapidly rotating the beam. Sediment particles in the water reflect this beam, and the chord length of these particles is calculated from the reflection time. The volume of sediment in the water can be calculated from the measurement results [8,9,10,11,12].

Acoustic methods for measuring the SSC or turbidity in water employ sound waves that penetrate water. The intensity of the acoustic backscatter from suspended sediment is utilized to estimate the SSC or turbidity. In the context of measuring the SSC in water using the acoustic method, several researchers have conducted laboratory experiments, employing water tanks and introducing small solid particles to water, such as glass beads or kaolin, to simulate scattering substances. These experiments have involved deploying relevant instruments, including optical and laser diffraction methods, or have directly sampled environmental materials. Several scholars have conducted laboratory validations of specific techniques in this research area, as referenced in [13,14,15,16,17,18,19,20,21,22,23,24,25]. A prominent example is the work of Deines [26], who conducted laboratory experiments and derived equations linking the SSC to the intensity of the sound echo received by an acoustic Doppler current profiler (ADCP) in water.

Additionally, some researchers have explored the feasibility of measuring the SSC in natural water bodies by using acoustic instruments through on-site observations. The studies conducted by [15,27,28,29,30,31,32,33,34,35,36,37,38,39,40,41,42,43] involved fixed-point observations in lakes, oceans, and estuaries. These studies utilized acoustic instruments in conjunction with other equipment to obtain the SSC. By calibrating and comparing data obtained through acoustic methods with those obtained through traditional approaches, the aforementioned studies demonstrated the reliability of acoustic instruments in accurately measuring SSC values within specific ranges.

A major challenge in using acoustic instruments for detecting sediment particles is the interaction between sound waves and suspended sediment particles. This interaction complicates the prediction and control of particle behavior. Studies conducted by Ha et al. [15], Guerrero et al. [16], and Hoitink and Hoekstra [34] demonstrated that different sound waves react differently to particles of differing sizes, with the conditions favorable for observation when the sound wavelength is commensurate with the radius of the sediment particles, in accordance with the principles of Rayleigh scattering.

To address potential inaccuracies in SSC estimations made from single-frequency sound pulses, researchers such as [16,22,24,27,30,32,39] have employed multiple sound frequency pulses in their experiments. This approach has enabled a better understanding of the advantages and error considerations associated with the utilization of multi-frequency sound waves. In addition, Thorne and Hanes [13] and Guerrero et al. [16] emphasized the strong influence of waterborne bubbles on acoustic observations in aquatic environments. Accounting for these considerations is imperative when employing acoustic techniques.

Relevant studies on measuring SSC in natural water bodies using acoustic instruments have primarily focused on relatively shallow waters. However, the studies cited in [5,33,34,37,39,42,43] conducted experiments in deeper waters. Specifically, Hawley [33], Park and Lee [39], and Wang et al. [43] used bottom-fixed equipment for their experiments, while Hoitink and Hoekstra [34], Wang and Gao [37], and Avila et al. [42] applied a post-sampling calibration technique.

Previous studies have developed various methods to measure SSC or turbidity in water. However, these methods have not been widely applied to acquire long-term, real-time data in coastal waters. Additionally, the relationships between SSC or turbidity and environmental factors such as rainfall, tides, currents, and waves have been insufficiently explored. Gaining a better understanding of how SSC or turbidity correlates with these factors could provide valuable insights into coastal dynamics and changes in water quality. Such knowledge is also critical for selecting appropriate sites for desalination plants.

In this study, an acoustic Doppler current profiler (ADCP) was installed on an ocean data buoy to obtain real-time and long-term sound echo intensity data. Simultaneously, a turbidimeter was deployed on the mooring line of the buoy to measure water turbidity. By establishing a correlation between sound echo intensity and water turbidity, real-time echo intensities collected by the ADCP can be converted into turbidity values. Because ADCP can provide an echo intensity profile, converting this into turbidity will yield a turbidity profile. This offers a significant advantage over other methods that generally measure turbidity at a single water depth. On-site experiments conducted from 6 June to 16 August 2017 in the waters of Qigu, Taiwan, yielded promising results. The correlations between water turbidity recorded during this period and rainfall, tides, currents, and waves were then elucidated.

This paper commences with a comprehensive review of relevant theories, covering the active sonar equation utilized by the ADCP and the physical mechanism underlying the measurement of sound waves for sediment particles suspended in water. Section 3 outlines the experimental setup employed in this study, detailing the installation of an ADCP and a turbidimeter on a buoy. Section 4 details the calibration process, which involved correlating the backscatter intensity measured by the ADCP with turbidity measurements. Section 5 provides a detailed discussion of the correlation between turbidity and other environmental factors such as rainfall, tide level, current velocity, and ocean waves. Finally, variation in the sound echo intensity measured by the ADCP with respect to water depth is discussed.

## 2. Related Theories

### 2.1. Properties and Phenomena of Estuary Sediment

Rivers that transport sediment to the ocean undergo substantial changes due to considerable variations in factors such as salinity and concentration, resulting in complex phenomena such as diffusion, mixing, and plume formation [44].

Wright and Nittrouer [44] identified four distinct processes involved in the transport of sediment from rivers to the ocean, as illustrated in Figure 1. The first process, termed “supply via plumes”, involves sediment’s entry into the ocean through river outflows. The density difference between freshwater and seawater plays a crucial role and results in pronounced vertical stratification. Two types of plumes are identified: the hypopycnal plume, characterized by lower-density river water floating atop seawater, forming a tongue-like distribution that transports sediment away from the estuary for dispersion, and the hyperpycnal plume, which occurs when river water is denser, causing it to sink below the seawater and rapidly spread along the seafloor.

The second process, called “initial deposition”, is influenced by factors such as the quantity and size of suspended particles, their settling velocity, their cohesiveness, and the density and viscosity of the water. These factors influence the ability of sediment to accumulate along the shoreline. The third process, known as “resuspension and transport”, occurs once the sediment has settled. This stage involves the resuspension and movement of suspended sediment due to waves, currents, and other factors that create bed shear stress. The fourth process, known as “long-term net accumulation”, involves the burial of sediment beneath the mixed layer, resulting in its gradual accumulation over time.

### 2.2. Underwater Sound Wave Transmission

#### 2.2.1. Equation for Active Sonar

The active sonar equation in underwater acoustics describes the situation in which the received echo signal surpasses a predefined detection threshold, as formulated in Equation (1) [45]. This equation considers the nonisotropic emission of sound waves from the transmitter along the axis of the sound beam to the target, the bidirectional transmission losses experienced during the round-trip process, and the influence of environmental noise.
(1)(SL−2·TL+TS)−(NL−AG)=DT

In this equation, SL (source level) reflects the intensity of sound emitted by the source. TL, the transmission loss, expresses the attenuation of sound energy during propagation. TS, the target strength, characterizes the reflection characteristics of the target. NL represents the ambient noise level, signifying the background noise in the environment. AG represents the array gain, which is the degree of amplification of the received signal by the sensor array. Finally, DT represents the detection threshold, that is, the minimum signal level required for successful detection.

#### 2.2.2. Attenuation and Absorption of Sound Waves in Water

Propagation of sound waves in water involves both attenuation and absorption, which contribute to the total transmission loss in the active sonar equation. This can be expressed in the following equation:(2)$$TL_{total}=TL_{attenuation}+TL_{absorption}$$
where $TL_{total}$ represents the total transmission loss (dB/m), $TL_{attenuation}$ represents the loss due to attenuation (dB/m), and $TL_{absorption}$ represents the loss due to absorption (dB/m). Attenuation involves the geometric spreading of energy as sound waves propagate without any associated energy loss. Absorption occurs when sound waves interact with the water medium or impurities within it (e.g., salt substances and suspended sediment), resulting in a decrease in energy [45].

#### 2.2.3. Rayleigh Scattering Theory

The interaction between fixed single-frequency sound emitted by underwater acoustic instruments (e.g., ADCPs) and suspended sediment particles is best explained through Rayleigh scattering. According to the Rayleigh scattering model, as proposed by Reichel and Nachtnebel [46], the following condition is applicable:(3)0.1<2·π·a/λ<1

Here, a is the radius of the sediment particles (m), and λ is the wavelength of the sound waves (m). When the condition for Rayleigh scattering is met, the measurement resolution is high. Therefore, according to the Rayleigh scattering condition, an ADCP operating at a frequency of 600 kHz can effectively measure sediment particles with radii ranging from 40 to 400 μm.

#### 2.2.4. Active Sonar Equation Used by the RDI ADCP

Deines [26] derived an equation to describe the relationship between the intensity of a sound echo originating from the RDI ADCP (Subsea Technology & Rentals, Aberdeen, United Kingdom) and the SSC:(4)$$\begin{array}{l} 10\cdot \log_{10}(SSC)=C+10\cdot \log_{10}[(T+273.16)\cdot R^{2}]-L_{DBM}\\ -P_{DBW}+2\cdot \alpha \cdot R+K_{c}\cdot (E-E_{r}) \end{array}$$
where SSC represents the suspended sediment concentration (mg/L); C is a combined parameter (dB); T is temperature of the transducer (°C); R is the range along the beam (slant range) to the scatterers (m); $L_{DBM}$ is a specific parameter related to the sound wave wavelength of the ADCP instrument (dB); $P_{DBW}$ is a specific parameter related to the power of the ADCP instrument (dB); α is the absorption coefficient of water (dB/m); $K_{c}$ is the receiver indicator, which is a proportionality factor (dB/counts); E is the received echo intensity of the ADCP (counts); and $E_{r}$ is the background echo intensity (counts).

Assuming that factors such as the ADCP’s position, water depth, temperature, sound wave wavelength, instrument power, absorption coefficient, and background echo intensity are constant, a composite parameter $C_{k}$ can be introduced. Equation (4) can subsequently be reformulated as follows:(5)$$10\cdot \log_{10}(SSC)=C_{k}+K_{c}\cdot E$$

By calibrating the unknown parameters $K_{c}$ and $C_{k}$ in Equation (5), the ADCP can effectively estimate the SSC from the intensity of its measured sound echo. Consequently, the following equation can be formulated:(6)$$SSC=10^{\left(\frac{C_{k}+K_{c}\cdot E}{10}\right)}$$

## 3. Experimental Setup and Water Sample Collection

### 3.1. Ocean Data Buoy

In this study, an ocean data buoy was deployed in the Qigu waters of Taiwan at 23°05′44.0″ N latitude and 120°00′30.0″ E longitude (see Figure 2). This location is about 2.7 km from the coastline of Tainan City, Taiwan, near the Zengwun River estuary. The buoy is approximately 8 km away from the estuary. The water depth at this site is approximately 18 m. A downward-looking RDI ADCP, operating at a frequency of 600 kHz, was affixed to the buoy’s pedestal. The primary function of the ADCP was to measure ocean currents and their profiles. However, it had capabilities other than measuring ocean current velocity and direction; it also measured sound echo intensity, which can be correlated with SSC or turbidity. The ADCP records echo intensity in a counts format, which can then be converted into decibels (dB) for evaluating the transmission loss. Measurements were made from 16:55 on 6 June 2017 to 05:59 on 16 August 2017. For ease of analysis, the echo intensity data were retained in the counts format. The observational bins of the ADCP had 1 m intervals, with the first bin located at a depth of 1.5 m. Additionally, an accelerometer–tilt–compass (ATC) sensor was installed on the buoy to capture data on ocean wave activity.

The data buoy was equipped with an independent power system consisting of three solar panels and six storage batteries to support long-term monitoring. It was also fitted with a data logger and a General Packet Radio Service (GPRS) module. This configuration enabled both the RDI ADCP and the ATC sensor to connect to the data logger, allowing real-time transmission of collected data to land-based users via 4G mobile networks or satellite communication. Consequently, if turbidity or SSC could be inferred from the sound echo intensity measured by the ADCP, these values could be remotely monitored and accessed in real time. Additionally, since the ADCP provides profiles of sound echo intensity, real-time profiles of turbidity or SSC could also be obtained.

In addition, a YSI EXO2 Multi-Parameter Water Quality Sonde, hereafter referred to as YSI EXO2 (Xylem, Yellow Springs, OH, USA), outfitted with an optical turbidity sensor, was employed to measure water turbidity. The YSI EXO2 was affixed to the mooring line of the buoy at a submerged depth of 13 m and operated from 16:00 on 6 June 2017 to 05:45 on 21 June 2017. As a self-recording instrument, the YSI EXO2 had limited ability for real-time data transmission.

On 16 August 2017, water samples were gathered at the study site for laboratory analysis. Three replicates were obtained at each of the following five depths: 3.0, 4.5, 6.5, 7.5, and 9.5 m. Additionally, a sediment sample was collected from the seabed. Figure 3 illustrates the deployment of the ocean data buoy for monitoring sound echo intensity, a turbidimeter for measuring turbidity, and the setup for collecting water samples in the Qigu waters. During sampling, strong currents may have caused the rope attached to the sampling bottle to deviate from its vertical alignment. Based on a rough estimation, the slant angle could reach 40–50 degrees. Consequently, the actual depths at which measurements were made may have differed from the designated depths.

### 3.2. Turbidity and Concentration of Water Samples

Fifteen water samples were collected in the field at five depths (3.0–9.5 m). The turbidity of each sample was determined using a HACH 2100N Laboratory Turbidimeter (Hach, Loveland, CO, USA). Five replicates were taken for each sample, and their turbidity was averaged. In addition, the water samples were filtered, dried, and weighed in the laboratory to determine their SSC.

Figure 4a,b depict the variations in average turbidity and concentration with respect to water depth, respectively. Figure 4a reveals a marginal decrease in turbidity with increasing water depth, but the change was not significant, likely due to the consistently low turbidity levels measured throughout the water column. Similarly, the concentration values in Figure 4b decreased with increasing water depth, although this trend was neither pronounced nor consistent. Notably, the water sampling process coincided with the ebb tide, as indicated by tidal data provided by the Water Resources Agency, Ministry of Economic Affairs, Taiwan.

### 3.3. Seabed Sediment Characteristics at the Study Site

Sediment from the seabed near the Qigu Buoy—collected on 16 August 2017—was analyzed to determine its particle size distribution. The analysis was carried out at the Center for Micro/Nano Science and Technology, National Cheng Kung University, using a DelsaNano C (Beckman Coulter, Inc., Brea, CA, USA). This instrument employs photon correlation spectroscopy to measure the size distribution of particles in a liquid medium, with a measurement range of 0.6 nm to 7 μm. Therefore, particle sizes exceeding 7 μm or smaller than 0.6 nm may result in inaccurate measurements. The results, depicted in Figure 5 and Table 1, revealed a bimodal distribution with peaks at approximately 2.7 and 120 μm. This distribution indicated a diverse range of particle sizes, from coarse clay to fine sand. According to Rayleigh scattering theory (Section 2.2.3), the ADCP utilized in this study is most effective for particles with radii between 40 μm and 400 μm. The resolution for particles smaller than 40 μm may thus have been relatively inaccurate.

## 4. Calibration Between Sound Echo Intensity and Turbidity

### 4.1. Calibration of Water Sample Turbidity Versus Concentration

SSC and turbidity are both crucial indicators for assessing water quality, though they are measured in different units. Turbidity values are obtained using an optical turbidimeter, in this case, the HACH 2100N Laboratory Turbidimeter, which reports values in nephelometric turbidity units (NTUs). On the other hand, SSC values are determined through analytical tests that involve filtration, drying, and weighing and are typically expressed in milligrams per liter (mg/L).

The turbidity and SSC values obtained for each water sample, as discussed in Section 3.2, were calibrated (Figure 6). However, the correlation coefficient derived from linear regression analysis was only 0.0851, indicating a weak correlation. This unsatisfactory result was likely caused by the presence of translucent substances in the collected water samples, which may have led to inaccuracies in the optical turbidity measurements. The presence of such substances is possibly related to the active fishing industry in the Qigu waters. Additionally, the limited number of water samples could have contributed to the weak correlation between turbidity and SSC.

### 4.2. Calibration of Turbidity and Concentration of Kaolin Solution

Due to the low correlation between the turbidity measurements from the HACH 2100N Laboratory Turbidimeter and the SSC of water samples collected at the study site, a separate calibration test was conducted using kaolin solutions. These solutions had concentrations ranging from 10 to 180 mg/L, increasing in 10 mg/L increments. The objective of the calibration was to evaluate the relationship between turbidity, as measured by the YSI EXO2 Turbidimeter (the same model used on the buoy), and the actual concentration of suspended particles. The results, shown in Figure 7, revealed a high correlation coefficient of approximately 0.9974, indicating a strong relationship between the turbidity readings from the YSI EXO2 and the corresponding concentration values.

The complex environmental conditions in the Qigu waters, as indicated by the experimental findings in Section 4.1, led to significant discrepancies in turbidity-concentration calibration between the field water samples and the laboratory-calibrated kaolin solutions. As a result, it was decided to rely primarily on turbidity values in the subsequent analysis rather than converting them into concentration values.

### 4.3. Calibration of Backscatter Intensity from ADCP with Turbidity from YSI EXO2

The average echo intensity from the four acoustic transducers of the ADCP and the turbidity values measured by the YSI EXO2 were calibrated using Equation (5). Because the ADCP and the turbidimeter were operated over different periods, the data selected for calibration covered the period from 6 June to 21 June 2017, matching the measuring period of the YSI EXO2. Assuming a linear relationship between SSC and turbidity value, as indicated by the results in Figure 7, the term “SSC” in Equation (5) was directly substituted with the turbidity values (in NTU) from the YSI EXO2.

Figure 8 presents the results of calibration for the sound echo intensity determined by the ADCP versus the turbidity measured using the YSI EXO2. The solid line, expressed as y=0.2077x−17.434 [where x represents echo intensity and y=10·log (Turbidity in NTU)], is a linear regression line. The upper and lower trend lines, indicating one standard deviation above and below the linear regression line, are expressed as $y_{1}=0.2077x_{1}-13.294$ and $y_{2}=0.2077x_{2}-21.574$, respectively. The calibration results suggest a correlation coefficient of approximately 0.5.

Several studies, such as those by Guerrero et al. [16], Hoitink and Hoekstra [34], Park and Lee [39], and Bartholomä et al. [47], have conducted similar calibration tests between ADCP sound echo intensity and water turbidity measured using various optical sensors. These studies have typically obtained modest correlation coefficients and patterns similar to those illustrated in Figure 8.

The modest correlation coefficient obtained in this study may be attributable to the acoustic wave frequency of the ADCP. According to Equation (3), an ADCP operating at 600 kHz is ideally suited for Rayleigh scattering with particles having radii between 40 and 400 μm. As detailed in Section 3.3, the particle size distribution in the Qigu waters had bimodal pattern peaks at approximately 2.7 and 120 μm. Consequently, 600 kHz sound waves may not have been effectively scattered by particles of size near 2.7 μm. Conversely, the optical method employed by the YSI EXO2 may have overestimated the turbidity in the Qigu waters, possibly due to the semitransparent nature of certain nonsediment particles. An improvement could be made by using acoustic instruments with multiple operating frequencies to better detect particles smaller than 40 μm.

Despite this relatively modest correlation, the results for the calibration between the ADCP’s sound echo intensity and the YSI EXO2’s turbidity measurements exhibited a distinct band-like distribution pattern. Notably, most of the calibration data points fell within one standard deviation above and below the established linear regression line. In Equation (5), certain parameters, such as the water depth, temperature, and water absorption coefficient, were assumed to be fixed. However, the YSI EXO2, being attached to the buoy’s mooring chain, experienced vertical movement due to wave motion, resulting in deviations from the depth of 13 m. These variations in several of the parameters likely contributed to the observed instability in the intercept of the calibration results, leading to the distinct band-like distribution in the calibration outcomes.

On the basis of the calibrated results presented in Figure 8, the parameters $K_{c}$ and $C_{k}$ in Equation (5) were determined to be $K_{c}=0.2077$ and $C_{k}=-17.434$. Substituting these values into Equation (6) and substituting “SSC” with “Turbidity” (measured in NTUs) yielded the following equation:(7)$$Turbidity=10^{\left(\frac{-17.434+0.2077\cdot E}{10}\right)}$$

Equation (7) indicates that monitoring the echo intensity value E by using the ADCP at a given time allows for the conversion of this value into a corresponding turbidity value.

Notably, in Equation (5) or (7), the determination of the turbidity from the echo intensity measured by the ADCP relies on two key parameters, $K_{c}$ and $C_{k}$. However, the calibration results presented in Figure 8 indicate a distinct band-like distribution pattern with a modest correlation coefficient of approximately 0.5. To assess the viability of using the ADCP-obtained echo intensity to determine the turbidity, another calibration relationship was established. This new relationship used the same echo intensity and turbidity values as those in Figure 8 but focused on data collected during the initial 10 days (6–15 June 2017) of the YSI EXO2’s measuring period rather than the entire period (6–21 June 2017). The results of this new calibration are presented in Figure 9. The updated calibration relationship was then employed to convert the echo intensity values into turbidity values for the period 16–21 June 2017. The results of this conversion were subsequently compared with the turbidity values obtained using the YSI EXO2.

The characteristics observed in Figure 9 resemble those in Figure 8, albeit with a slightly lower correlation coefficient of 0.4543. The linear regression line can be expressed as y=0.1819x−15.108, and the upper and lower linear trend lines, deviating by one standard deviation from the regression line, are represented by $y_{1}=0.1819x_{1}-11.469$ and $y_{2}=0.1819x_{2}-18.746$, respectively.

Utilizing the linear regression line with the equation y=0.1819x−15.108, the conversion of the echo intensity acquired over 6 consecutive days (16–21 June 2017) into turbidity values resulted in the outcomes presented in Figure 10a. Alternatively, adopting the upper linear trend line in Figure 9 ($y_{1}=0.1819x_{1}-11.469$) produced the outcomes depicted in Figure 10b. Figure 10a,b includes the measured turbidity values for comparison.

The results presented in Figure 10a indicate that during periods of low turbidity (16–17 June 2017), the estimated turbidity was close to the measured values. However, during periods of high turbidity (18–20 June), the estimated values were consistently lower than the measured values. Conversely, Figure 10b demonstrates that although the estimated turbidity values slightly exceeded the measured values during the initial 2 days, the correlation between the estimated and measured values was robust for the subsequent 3 days, both in overall trend and magnitude. Notably, certain turbidity values measured by the YSI EXO2 from 18 to 19 June were unusually large. The comparison presented in Figure 10a,b illustrates that for waters with higher turbidity, employing the upper trend line from Figure 9 to determine turbidity from the echo intensity yielded more accurate results than employing the linear regression line.

The upper trend line in Figure 8 can be expressed as $y_{1}=0.2077x_{1}-13.294$. Consequently, Equation (7)—for converting the echo intensity into turbidity, based on the observation data from 6 to 21 June 2017—was modified as follows:(8)$$Turbidity=10^{\left(\frac{-13.294+0.2077\cdot E}{10}\right)}$$

In practical applications, the accuracy of turbidity values may be more critical when turbidity is high than when it is low. Accordingly, this study preferred Equation (8) over Equation (7) for generating a time series of estimated turbidity values from the echo intensity data collected by the ADCP. The outcomes of estimated turbidity at a water depth of 13 m for the period 6 June to 16 August 2017, derived using Equation (8), are presented in Figure 11. Notably, the turbidity values obtained using Equation (8) were 2.594 (equivalent to the factor 10^0.414^) times higher than those values obtained using Equation (7).

## 5. Correlations Between Coastal Water Turbidity and Meteorological and Oceanographic Data

In this section, we explore the correlations between estimated turbidity values for the period 6 June to 16 August 2017, as detailed in Section 4.3, and various meteorological and oceanographic parameters such as rainfall, tides, currents, and waves. The variation in turbidity with respect to water depth is also discussed. As shown in Figure 1, any environmental factor that introduces sediment loads or affects water flow, leading to the resuspension of sediment, will affect turbidity levels. Consequently, factors such as rainfall, slope failure, tides, currents, waves, and wind all influence turbidity. In this study, we examined the relationships between turbidity and rainfall, tides, currents, and waves, which account for most of the aforementioned environmental factors.

### 5.1. Precipitation in the Qigu Region from 6 June to 16 August 2017

Rainfall data from the Qigu area were crucial for understanding the effects of post-rainfall surface runoff on the discharge of terrestrial sediment into the ocean. Rainfall records for the period 6 June to 16 August 2017 from the Qigu, Annan, Anding, and Xigang Rainfall Stations were gathered, courtesy of the Central Weather Administration, Taiwan. These stations are located near the Zengwun River estuary in Tainan City, Taiwan, with distances less than 18 km between each station and the estuary. The rainfall rates at these stations are represented in Figure 12 by columns with various stripe patterns.

An analysis of precipitation patterns unveiled a distinct “plum rain period” in June, traditionally linked to a local monsoon coinciding with the plum ripening period, particularly on 14–17 June 2017. In July and August, a significant amount of rainfall was recorded, particularly from 30 July to 2 August 2017; this rainfall was attributable to the influences of Typhoon Nesat and Typhoon Haitang. Typhoon Nesat approached the Qigu waters on 29 July 2017, with central wind speeds ranging from 32.8 to 50.7 m/s. Typhoon Haitang arrived on 30 July 2017, with central wind speeds varying between 17.5 and 32.7 m/s in the Qigu vicinity. On most other days, the rainfall was minimal. The specific characteristics of these two typhoons are detailed in Table 2.

From the rainfall data, three distinct temporal intervals were identified:The plum rain period from 14 June to 17 June is marked by considerable rainfall.The rainless period after the plum rain from 18 June to 20 June is marked by minimal rainfall immediately following the plum rain event.The typhoon period from 30 July to 1 August is defined by substantial typhoon-related precipitation.

### 5.2. Correlation Between Turbidity and Tide Level

This subsection examines the correlation between turbidity and tide levels during the plum rain period (14–17 June 2017), rainless post-plum-rain period (18–20 June 2017), and typhoon period (30 July–1 August 2017). The corresponding findings are illustrated in Figure 13a–c.

Tidal data were obtained from the Sicao Tide Station, positioned at coordinates 23°01′25.0″ N, 120°06′43.0″ E, approximately 13.5 km southeast of the Qigu Buoy. This station is under the jurisdiction of the Water Resources Agency, Ministry of Economic Affairs, Taiwan, and uses the Taiwan Vertical Datum 2001 (TWVD2001) for standardization, with the mean sea level at Keelung Harbor serving as the zero-reference point.

Figure 13a shows that turbidity at a depth of 13 m was significantly higher during the ebb and flood tide periods compared to the slack tide periods. Notably, turbidity levels exceeded 40 NTU during the ebb tide from 02:00 to 06:00 on 14 June and during the flood tide from 20:00 on 14 June to 02:00 on 15 June. Figure 13b indicates that turbidity values recorded during this period were substantially higher than those observed during the plum rain period. This can be attributed to the precipitation during the plum rain period, which triggered sediment transport from the Zengwun River to the Qigu waters, affecting the subsequent rainless post-plum-rain period, as illustrated by the four sedimentary processes shown in Figure 1.

The results presented in Figure 13a,b indicate higher turbidity levels during ebb and flood tide periods than during other periods. For instance, the turbidity value was 79.49 NTU at approximately 04:00 on 19 June, immediately following the flood tide and during the transition to slack water. Within the subsequent 4 h, the turbidity decreased to approximately 10 NTU.

Figure 13c illustrates that the turbidity values during the typhoon period were lower than those during the plum rain period and rainless post-plum-rain period. This implied that the effect of the typhoon on the bottom-layer turbidity (at a depth of 13 m) was weaker than the influence of the plum rain system. However, the turbidity values still substantially increased during ebb and flood tides.

### 5.3. Correlation Between Turbidity and Current Velocity

Figure 14a–c illustrates the correlations between turbidity and current velocity during the three identified time frames. Current velocity was measured at a depth of 13 m by using the ADCP positioned in the eleventh bin, aligning with the turbidimeter’s depth. Figure 14a illustrates that during the plum rain period (14–17 June 2017), periods of higher current velocity corresponded with noticeably higher turbidity. Peaks in turbidity (>40 NTU) aligned with periods of maximum current velocity during the ebb tide period from 02:00 to 06:00 on 14 June and the flood tide period from 20:00 from 14 June to 02:00 on 15 June.

Consistent with the findings presented in Figure 14a, Figure 14b reveals that higher current velocity was associated with noticeably higher turbidity. Particularly noteworthy was the peak turbidity (79.49 NTU) that occurred at approximately 04:00 on 19 June, following the flood tide and during the transition to slack water. This peak in turbidity coincided with the peak in the current velocity. As the current velocity decreased, the turbidity also decreased.

During the typhoon period (Figure 14c), the trend of higher turbidity with higher current velocity was again evident, although the trend was less strong compared with that shown in Figure 14a,b, likely due to the predominantly lower turbidity values during this period.

The temporal variation in turbidity, as shown in Figure 13 and Figure 14, can be explained using the theoretical framework proposed by Wright and Nittrouer [44]. During the plum rain period, heavy rainfall transports terrestrial sediment to the Zengwun River estuary, leading to a significant rise in turbidity during the rainless period (18–20 June 2017). This corresponds to the initial “supply via plumes” phase of sediment dispersal in coastal waters, as depicted in Figure 1. The stronger tidal currents during both ebb and flood tides resuspend previously settled sediment, further increasing turbidity in the water column. This aligns with the “resuspension and transport” phase. These observations are consistent with findings from [30,34,36,37,47,48,49], which show a strong correlation between suspended sediment concentration (SSC) and the strength of tidal currents. These studies support the conclusions drawn from our results.

### 5.4. Correlation Between Turbidity and Wave Energy

This subsection investigates the effects of water waves on turbidity in relation to wave energy. An ATC was installed on the buoy to record wave data in the Qigu waters. The resultant data set encompassed data on significant wave height, the peak wave period, and the directional wave spectrum. The accuracy of the wave data acquired by the ATC sensor was validated by comparing these data with data obtained from a GNSS buoy, as outlined in Lin et al. [50,51]. The significant wave height and peak wave period recorded from 1 June to 31 August 2017 are illustrated in Figure 15.

Wave energy (W) can be calculated using the following equation proposed by McCormick [52]:(9)$$W=0.49\times H_{s}^{2}\times (0.9\times T_{p})$$
where W represents the wave energy (W/m), $H_{s}$ is the significant wave height (m), and $T_{p}$ is the peak period (s).

The relationship between turbidity and wave energy across the three observed time frames is illustrated in Figure 16a–c. In Figure 16a,b, wave energy remained below 10 kW/m, with turbidity fluctuating between 5 and 50 NTU in Figure 16a and between 5 and 80 NTU in Figure 16b. During the typhoon period [Figure 16c], the highest wave energy, approximately 114 kW/m, was recorded around 03:00 and 07:00 on 31 July 2017. However, this spike in wave energy did not coincide with a significant increase in turbidity.

The results shown in Figure 16a–c suggest no clear correlation between wave energy and turbidity at a depth of 13 m. This observation contrasts with findings from other studies, such as Bartholomä et al. [47] and Luo and Zhang [53], which suggest that waves can influence suspended sediment concentrations. The discrepancy may be due to differences in water depth at the respective study sites. Bartholomä et al. [47] conducted their research south of Spiekeroog in the German Wadden Sea, while Luo and Zhang [53] gathered data near an island in Lake Taihu, China—both in shallow waters. In contrast, the current study, conducted at a depth of approximately 13 m, suggests that wave-induced motion is unlikely to impact the seabed at this depth, and therefore, sediment resuspension due to wave action may not occur.

### 5.5. Variation in Echo Intensity with Respect to Water Depth

The turbidity in a water body can vary both horizontally and vertically. The RDI ADCP used in this study allowed for profiling both current velocity and echo intensity, enabling an analysis of turbidity at various water depths. As shown in Figure 3, the first bin of the ADCP, located at an approximate depth of 3.5 m, corresponded to the surface layer of the water column. The eighth bin, at around 10.5 m, represented the intermediate layer, and the fifteenth bin, at about 17.5 m, represented the bottom layer.

Figure 17 illustrates the temporal evolution of the echo intensities measured between 6 June and 14 August 2017 in these three distinct layers. During the study period, the echo intensities in the intermediate and bottom layers were nearly identical, while the intensity in the surface layer was higher.

During the study period, two consecutive typhoons, Nesat and Haitang, passed over the Qigu waters. As shown in Figure 17, during Typhoon Haitang (29 July to 1 August 2017), the echo intensities in all three layers increased significantly, with the most pronounced changes observed in the surface layer. The echo intensity in the surface layer on 30 July 2017 reached its highest value during the entire observation period (6 June to 14 August 2017), suggesting that wave activity may influence sound echo intensity or turbidity at a depth of 3.5 m.

Based on the calibration discussed in Section 4.3, where higher sound echo intensity was found to correspond with increased turbidity, the results in Figure 17 indicate that turbidity in the surface layer was higher than in the intermediate and bottom layers.

## 6. Conclusions

In this study, an ADCP and a self-recording turbidimeter were installed on an ocean data buoy deployed in the Qigu waters, Taiwan, to measure sound echo intensity and turbidity, respectively, and a correlation between these parameters was established. Additionally, water samples from the Qigu waters were collected for concentration analysis, and sediment near the Qigu Buoy was collected to determine the sediment’s particle size distribution.

Calibration of turbidity values against concentration values in water samples yielded a correlation coefficient of only 0.0851, indicating an unsatisfactory correlation. This outcome could be attributed to the complex environmental conditions present in the studied area or to the limited number of water samples collected.

Calibration between the ADCP-obtained echo intensity and turbidimeter-measured turbidity values at a water depth of 13 m revealed a correlation coefficient of approximately 0.5. The results indicated a banded distribution pattern, a result that aligns with findings from other researchers [16,34,39,47].

The correlation coefficient obtained in the present study was modest, perhaps because of the specific acoustic wave frequency used by the ADCP. An ADCP operating at 600 kHz is ideally suited for Rayleigh scattering of suspended particles with radii ranging from 40 to 400 μm. However, the particle size distribution in the Qigu waters was found to be bimodal, with peaks at approximately 2.7 and 120 μm. Consequently, sound waves at a frequency of 600 kHz may not have been effectively scattered by particles of size near 2.7 μm.

The echo intensity data obtained by the ADCP on the Qigu Buoy—collected from 6 June to 16 August 2017—were converted into a turbidity time series by using the established calibration formula. However, this calibration was conducted during the summer, when water temperatures were relatively higher. The effect of water temperature on the calibration formula needs to be clarified in future research. The correlations between ocean water turbidity and tide levels, currents, and waves were explored during three periods characterized by differing rainfall conditions: the plum rain period, the rainless post-plum-rain period, and the typhoon period.

Our results indicated considerable increases in turbidity during ebb and flood tide periods. Similarly, strong currents were discovered to increase turbidity. The influence of waves on turbidity was examined in relation to the waves’ energy. Our results indicated that wave action had minimal influence on turbidity at a water depth of 13 m, even during the typhoon period, likely due to the weak effect of waves at this depth. These results suggest that the tides and currents are the primary forces driving sediment transport in the Qigu waters.

Additionally, this study examined variations in echo intensity with respect to water depth. Our results demonstrated that the echo intensity for the surface layer (depth of 3.5 m) was higher than that in the intermediate and bottom layers (depths of 10.5 and 17.5 m, respectively). The echo intensities for the intermediate and bottom layers were found to be nearly identical.

Overall, this study provides a reliable and innovative technique for real-time and long-term monitoring of coastal water turbidity. By employing the ADCP for turbidity assessment, the proposed approach eliminates the need to place turbidity sensors in deep waters, thus mitigating the risks of instrument damage or loss.

## Figures and Tables

**Figure 1 sensors-24-06979-f001:**
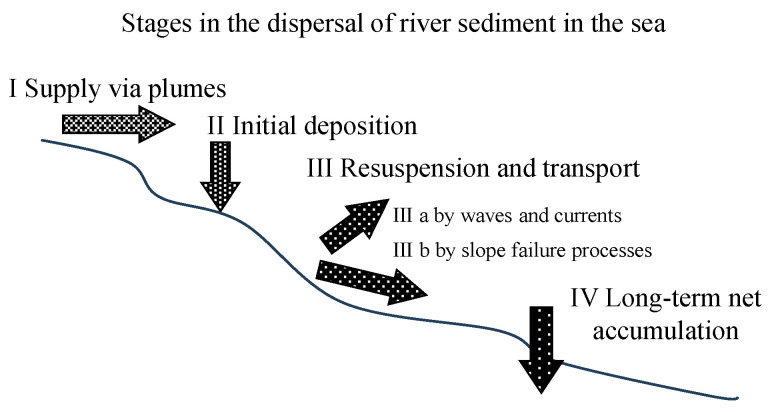
Conceptual illustration of the four primary stages of river sediment dispersal in the coastal ocean [44].

**Figure 2 sensors-24-06979-f002:**
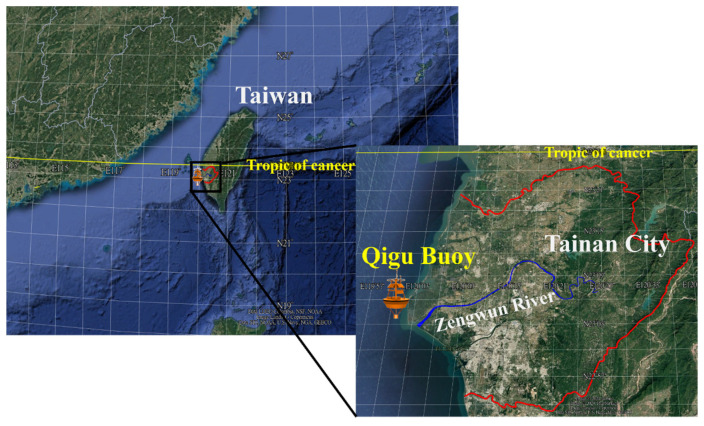
Geographical location of the Qigu ocean data buoy. (Color online).

**Figure 3 sensors-24-06979-f003:**
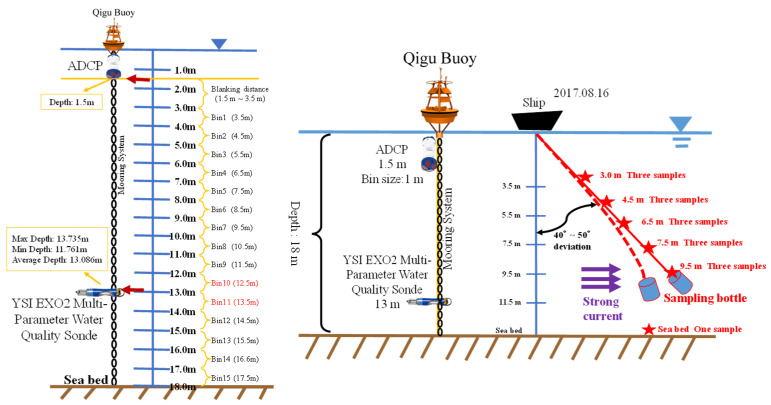
Deployment of an ocean data buoy for monitoring echo intensity, a turbidimeter for measuring turbidity, and the setup for water sample collection in the Qigu waters. The left figure illustrates the relative positions of the buoy, the ADCP, and the YSI EXO2 Multi-Parameter Water Quality Sonde. The right figure displays the depths at which water samples were gathered. (Color online).

**Figure 4 sensors-24-06979-f004:**
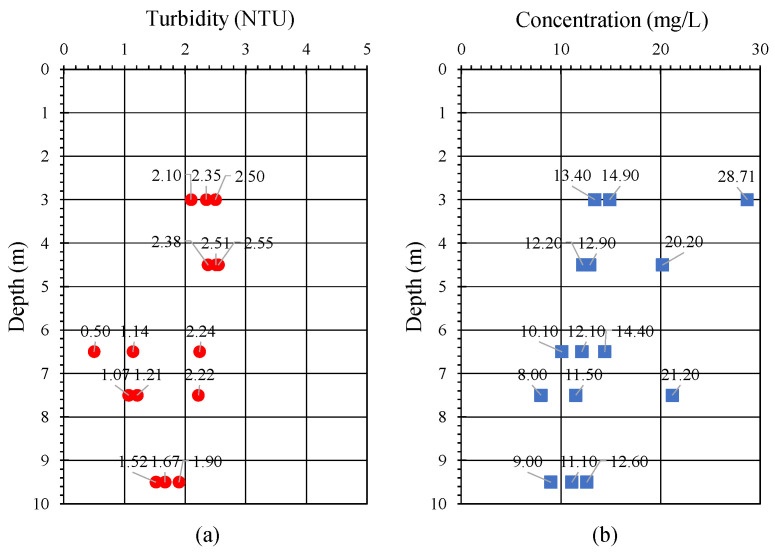
Variation in the (**a**) average turbidity (red dots) and (**b**) concentration (blue squares) of the water samples with respect to water depth. (Color online).

**Figure 5 sensors-24-06979-f005:**
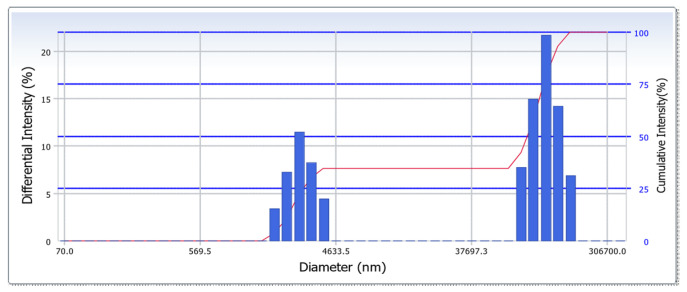
In situ sediment particle size distribution; the red line indicates the cumulative intensity of the particle sizes. (Color online).

**Figure 6 sensors-24-06979-f006:**
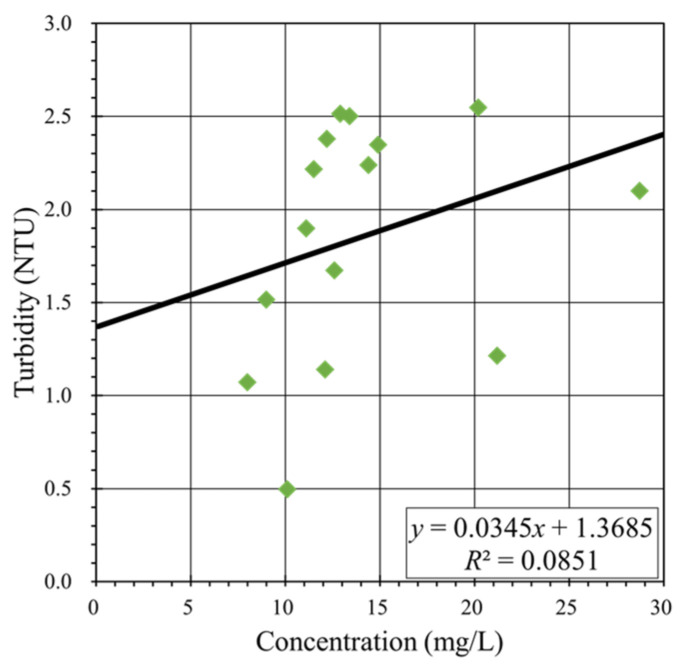
Calibration of turbidity versus concentration in on-site water samples. (Color online).

**Figure 7 sensors-24-06979-f007:**
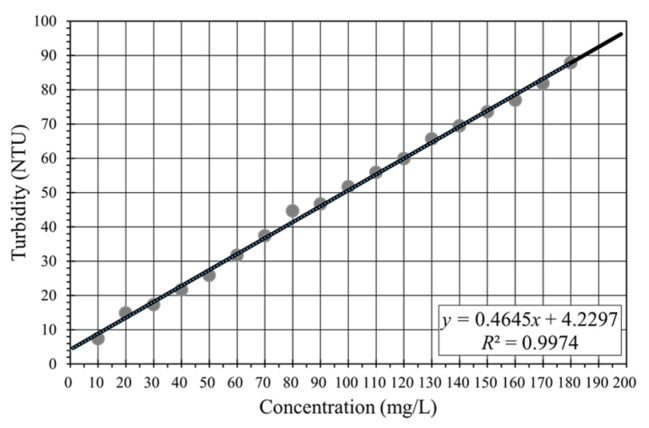
Calibration of turbidity versus concentration of kaolin solution.

**Figure 8 sensors-24-06979-f008:**
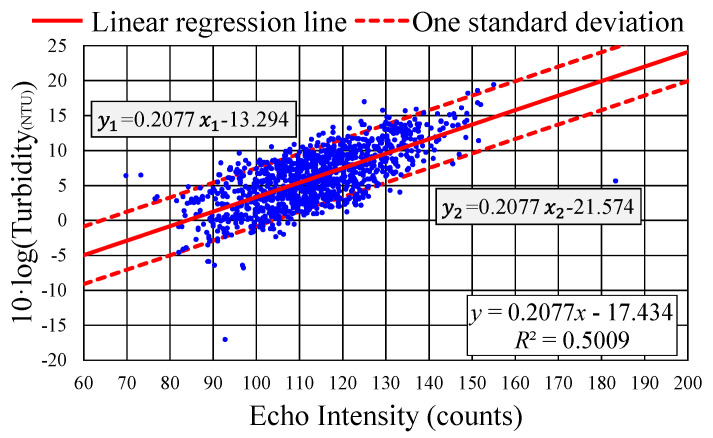
Calibration of turbidity versus sound echo intensity based on the data collected from 6 to 21 June 2017. (Color online).

**Figure 9 sensors-24-06979-f009:**
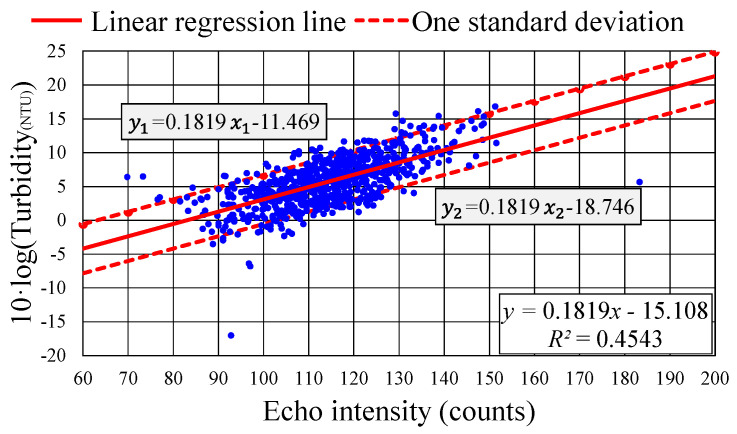
Calibration of turbidity versus sound echo intensity based on the data collected from 6 to 15 June 2017. (Color online).

**Figure 10 sensors-24-06979-f010:**
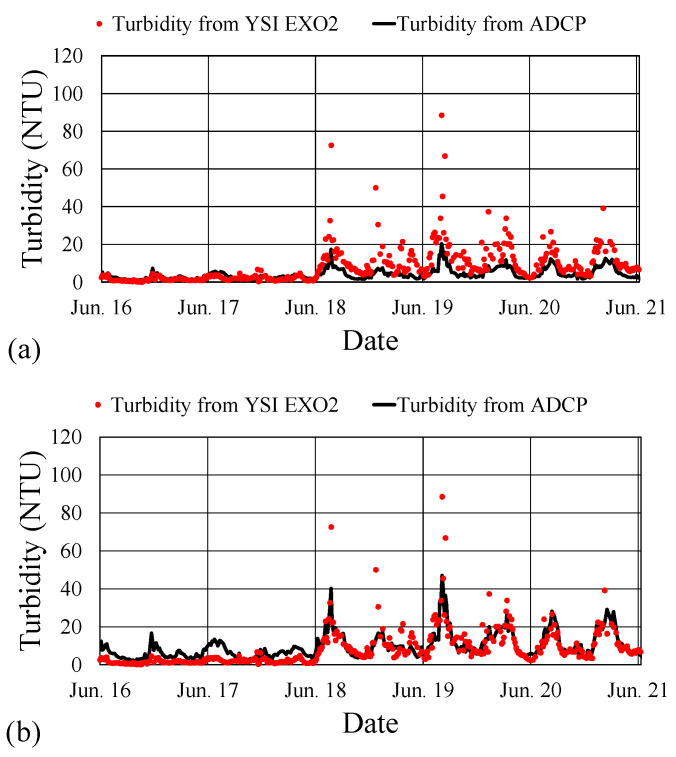
Comparison of estimated and measured turbidity values from 16 to 21 June 2017. The estimated values were obtained on the basis of (**a**) the linear regression line and (**b**) the upper trend line displayed in Figure 9. (Color online).

**Figure 11 sensors-24-06979-f011:**
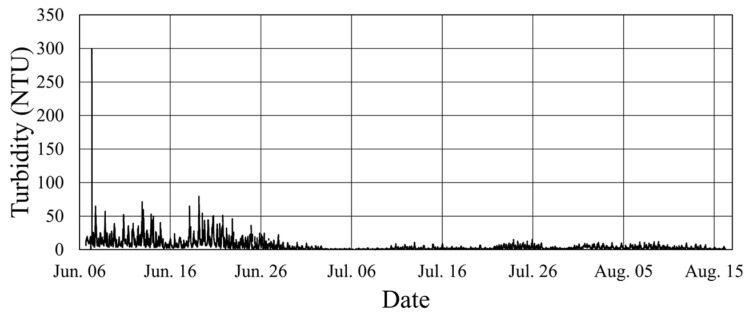
The time series of turbidity values estimated using Equation (8) for a water depth of 13 m and the period 6 June to 16 August 2017, based on the echo intensity values measured by the ADCP.

**Figure 12 sensors-24-06979-f012:**
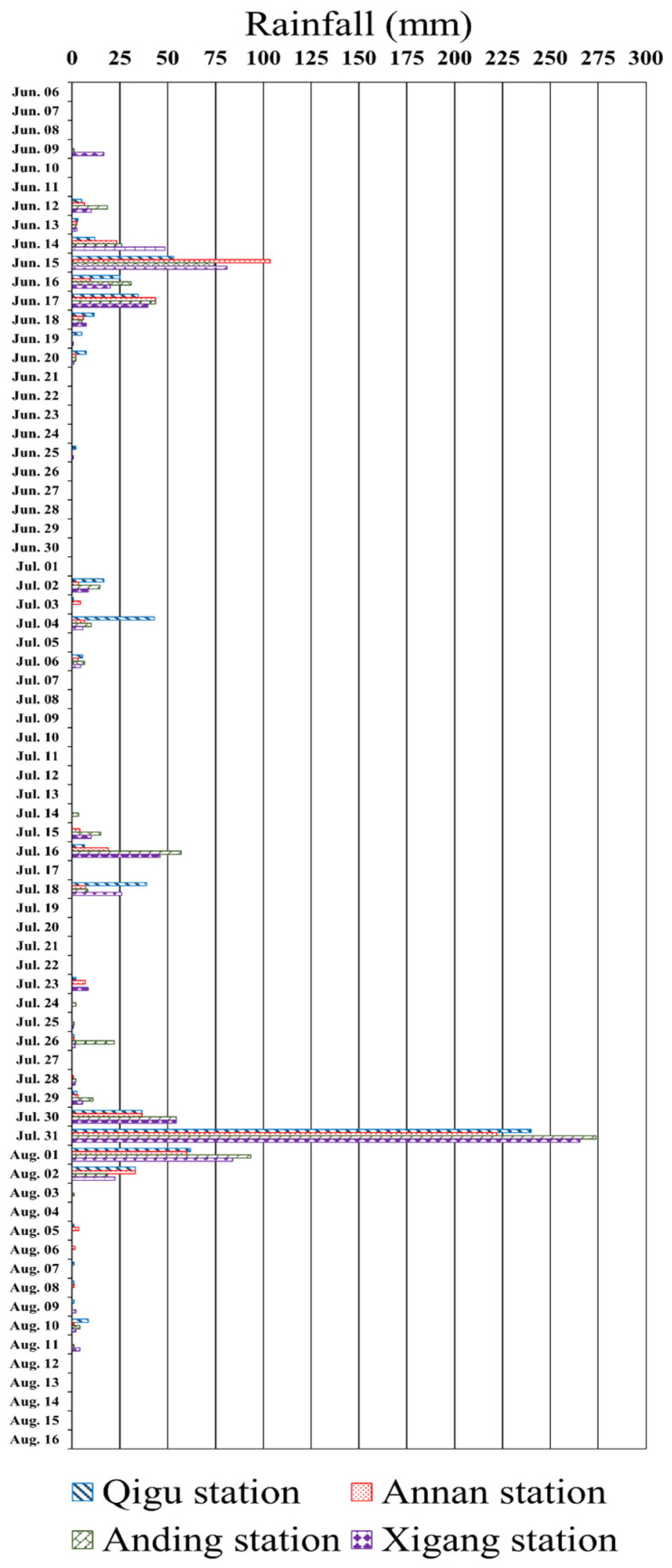
(color online) Rainfall rates in the Qigu region from 6 June to 16 August 2017 (accessed on 24 May 2018; data source: https://www.cwa.gov.tw/V8/C/).

**Figure 13 sensors-24-06979-f013:**
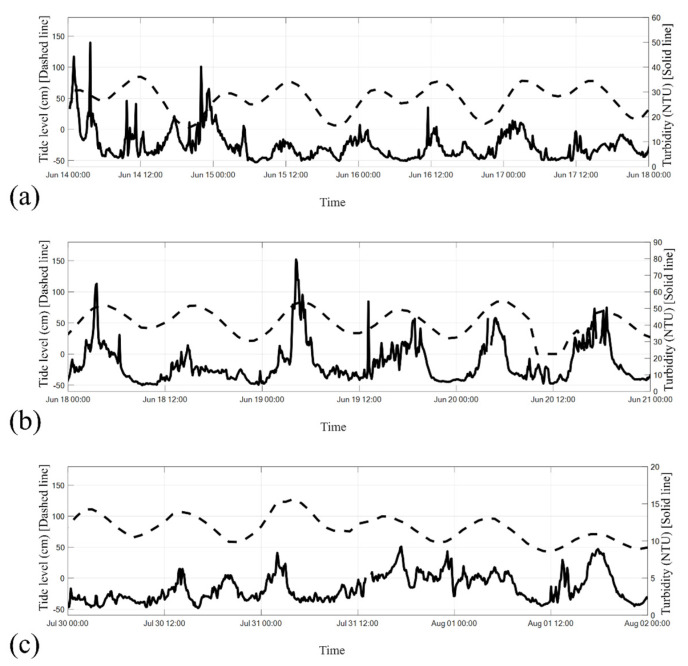
Correlation between turbidity values at a water depth of 13 m and tide levels recorded at the Sicao Tide Station: during the (**a**) plum rain period, (**b**) rainless post-plum-rain period, and (**c**) typhoon period. The dashed line represents the tide level, whereas the solid line represents turbidity.

**Figure 14 sensors-24-06979-f014:**
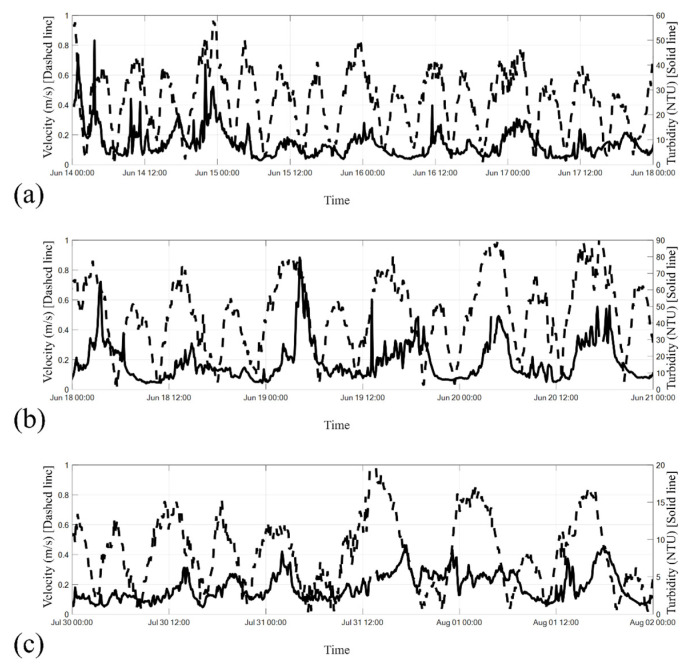
Correlation between the turbidity and current velocity at a water depth of 13 m during the (**a**) plum rain period, (**b**) rainless post-plum-rain period, and (**c**) typhoon period. The solid line represents turbidity, whereas the dashed line represents current velocity.

**Figure 15 sensors-24-06979-f015:**
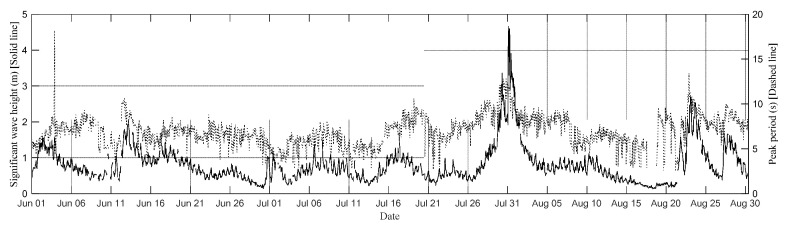
Significant wave height and peak wave period measured by the Qigu Buoy from 1 June to 31 August 2017 (solid line: significant wave height; dashed line: peak wave period).

**Figure 16 sensors-24-06979-f016:**
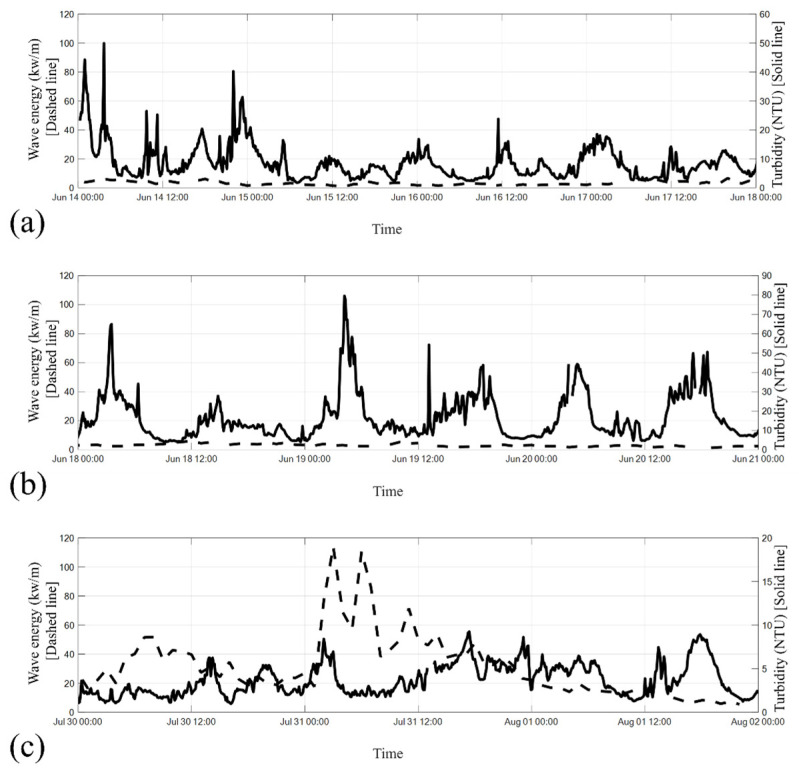
Correlation between the turbidity at a water depth of 13 m and wave energy at the location of the Qigu Buoy during the (**a**) plum rain period, (**b**) rainless post-plum-rain period, and (**c**) typhoon period. The solid line represents turbidity, and the dashed line represents wave energy.

**Figure 17 sensors-24-06979-f017:**
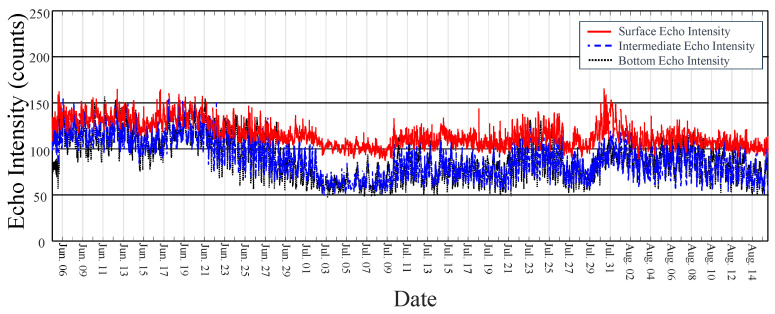
Echo intensity recorded by the ADCP from 6 June to 14 August 2017 at various water depths. The solid red line, dashed blue line, and dotted black line, respectively, denote the data for the surface, intermediate, and bottom layers. (Color online).

**Table 1 sensors-24-06979-t001:** Particle size distribution of seabed sediment from near Qigu Buoy.

Distribution Results		
Peak	Diameter (nm)	Standard Deviation
1	2724.6	598.3
2	120,293.7	26,592.1
Average	79,506.9	59,947.0

**Table 2 sensors-24-06979-t002:** Typhoon events during the study period (accessed on 24 May 2018; data source: https://rdc28.cwa.gov.tw/).

Year	2017	2017
Typhoon Serial No.	201709	201710
Typhoon Name	Nesat	Haitang
Typhoon Life Cycle	From 26 July 2017, 06:00:00 to 30 July 2017, 12:00:00	From 29 July 2017, 09:00:00 to 31 July 2017, 06:00:00
Minimum Pressure Near Typhoon Center (hPa)	955	990
Maximum Wind Speed Near Typhoon Center (m/s)	40	20

## Data Availability

Data are contained within the article.

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
