# Peer review of "Real-Time and Long-Term Monitoring of Coastal Water Turbidity Using an Ocean Buoy Equipped with an ADCP"

_sensors, 2024, doi:10.3390/s24216979_

Round 1

Reviewer 1 Report

Comments and Suggestions for Authors

In this paper, the relationship between acoustic echo intensity measured by ADCP and turbidity measured by turbidimeter is studied. The echo intensity data is converted into turbidity by this relationship. The real-time and long-term detection of turbidity in coastal waters is achieved through the deployment of buoys equipped with ADCP. The correlation between turbidity of coastal waters and marine environment is also studied, and illustrated by the measured data. The research content is innovative and practical.

        There are specific questions for the author to consider:

        (I) In Section 4.3, the paper argues that formula (8) is used instead of formula (7) to obtain more accurate turbidity calculation results when turbidity is high. What is the root cause behind this? Why is the calculation result of formula (7) not accurate enough in the case of high turbidity?

        (II) In Section 5, the relationship between turbidity and environment is studied. The research content of this paper is mainly to calculate turbidity through the echo intensity of ADCP, and the content of section 5 should be more focused on the relationship between the accuracy of turbidity calculation and the environment. It is suggested to add the error curve between the turbidity calculated by the echo intensity of ADCP and the turbidity measured by the turbidity meter, and the relationship between the error and the environment.

        (III) The experiments in the paper took place from June to August, when sea temperatures are higher in summer. If the experiment is conducted in winter, will the water temperature affect the calculation accuracy of turbidity value? It is recommended to add temperature-related content.

Comments on the Quality of English Language

none

Reviewer 2 Report

Comments and Suggestions for Authors

1.       The manuscript has limited innovation. As the author mentioned, the method of using ADCP to measure seawater turbidity has been in practice for over a decade or even several decades. This paper does not demonstrate significant innovation or breakthroughs in hardware development or data processing.

2.      The research does not reflect or emphasize the advantages of the ADCP measurement method in coastal turbidity monitoring.

3.     Lines 95 and 414: What additional environmental factors could influence turbidity levels?

4.     What potential interferences in the coastal ocean environment might compromise the detection levels and accuracy of measurements?

5.     Line 221: Why was a depth of 13 meters chosen for the Acoustic Doppler Current Profiler (ADCP) and turbidity sensor?

6.     Figure 5: The x-axis of this image is difficult to interpret. It is advisable to use clearer representations, such as 10, 100, 1,000, and 10,000 nm.

7.     What instruments and methods were employed to measure the particle size of the sediments? Given that the instrument is known to be inaccurate for particles smaller than 40 micrometers, the observed peak distribution at 2.7 micrometers is questionable, thereby rendering the entire distribution chart less meaningful.

8.     The discussion in subsection 3.3 regarding sediment characteristics lacks substantial analysis and conclusions. It should be rephrased to address the correlations between turbidity and the diameter distribution of sediments (not just suspended particles).

9.     Line 278 states that translucent substances may lead to inaccuracies in optical turbidity measurements. Why was the decision made to rely on turbidity values as noted in Line 301?

10.   Line 436 or Figure 12: The cumulative bar chart data from different stations are challenging to read. The author could provide separate listings or visual representations of rainfall intensity at each station.

11.     Section 5 provides data on rainfall, tides, wave energy, and other factors over time but does not delve into how these hydrological records affect the measured turbidity. What mechanisms drive the changes in turbidity?
